# Health Information Mistrust Is Directly Associated with Poor Sleep Quality: Evidence from a Population-Based Study

**DOI:** 10.3390/healthcare13121385

**Published:** 2025-06-10

**Authors:** Dietmar Ausserhofer, Christian J. Wiedermann, Verena Barbieri, Stefano Lombardo, Timon Gärtner, Klaus Eisendle, Giuliano Piccoliori, Adolf Engl

**Affiliations:** 1Institute of General Practice and Public Health, Claudiana—College of Health Professions, 39100 Bolzano, Italy; 2Claudiana Research, Claudiana—College of Health Professions, 39100 Bolzano, Italy; 3Provincial Institute for Statistics of the Autonomous Province of Bolzano-South Tyrol (ASTAT), 39100 Bolzano, Italy; 4Directorate, Claudiana—College of Health Professions, 39100 Bolzano, Italy

**Keywords:** sleep quality, mistrust, preventive health behavior, health information, public health

## Abstract

Background: Mistrust in professional health information may undermine population health by reducing engagement in preventive care and contributing to poorer health outcomes. Although sleep quality is a sensitive indicator of both psychosocial stress and health behavior, little is known about how mistrust influences sleep at the population level, and whether preventive health behavior mediates this relationship. Methods: A weighted cross-sectional analysis of a representative adult sample (*n* = 2090) from South Tyrol, Italy was conducted. Survey data included mistrust toward professional health information (Mistrust Index), five preventive health behaviors (Health Behavior Checklist, HBC), and sleep quality (Brief Pittsburgh Sleep Quality Index, B-PSQI). Associations between mistrust, behavior, and sleep were examined using multivariable linear regression, robust regression (Huber’s M-estimator), and nonparametric correlation. Results: Sociodemographic characteristics were not significantly associated with mistrust when weighted data were applied. Higher mistrust was associated with poorer sleep quality (β = 0.09, *p* = 0.003). Preventive health behaviors varied significantly across mistrust levels, with high-mistrust individuals less likely to report regular engagement (all *p* < 0.01). Regression analyses confirmed that mistrust was independently associated with poorer sleep quality, while preventive behaviors showed no significant relationship with sleep. Conclusions: Mistrust in professional health information is independently associated with poorer sleep quality and lower engagement in preventive behaviors. However, preventive behavior does not appear to mediate this relationship. These findings highlight mistrust as a direct and potentially modifiable risk factor for sleep disturbance at the population level.

## 1. Introduction

Sleep quality is increasingly recognized as a key determinant of adult health and well-being [1]. Poor sleep has been linked to a heightened risk of both physical and mental health disorders, and contributes significantly to reduced quality of life [2]. While traditional research has focused on biological and behavioral factors, recent public health investigations emphasize the role of psychosocial influences in outcomes [3]. In particular, a population’s trust in professional health information and their engagement with preventive healthcare services are emerging as important determinants [4].

Mistrust toward medical information encompasses a general skepticism toward professional health advice, public health recommendations, and healthcare institutions, and has been linked to reduced engagement with preventive services and poorer health outcomes [5]. Several studies have shown that such mistrust is widespread and associated with the lower use of preventive services, delayed diagnoses, and poorer health outcomes [6,7]. Individuals with heightened mistrust are less likely to attend preventive check-ups, follow treatment recommendations, or seek professional support for health concerns, including sleep disorders [8].

Preventive healthcare engagement, such as attending regular health screenings and vaccinations, is widely regarded as a cornerstone of population health [9]. The early detection and management of risk factors like hypertension, obesity, or stress can substantially improve sleep health [10]. Nonetheless, individuals who are distrustful of medical institutions tend to underutilize these preventive services [11], thereby missing opportunities to detect and manage sleep-related health risks.

In parallel, recent studies have highlighted possible cognitive and emotional pathways linking poor sleep to increased susceptibility to conspiracy beliefs [12,13]. Sleep deprivation may impair critical thinking, enhance emotional dysregulation, and increase the tendency to accept misinformation [14]. Although these findings open important avenues for understanding psychosocial influences on sleep, it remains unclear how general mistrust toward health information and preventive behavior differ in their mechanisms and impacts compared to conspiracy belief systems.

To conceptually ground the associations between mistrust in professional health information, preventive health behavior, and sleep quality, this study draws on the Health Belief Model (HBM). The HBM proposes that individual health behaviors are influenced by perceptions of susceptibility to illness, perceived severity of potential health threats, perceived benefits of action, and perceived barriers to action [15,16]. In the context of sleep health, mistrust may diminish perceived benefits of preventive measures and amplify perceived barriers, thereby leading to lower engagement in health-promoting behaviors and potentially poorer sleep outcomes (Figure 1). The Health Belief Model (HBM) serves as the conceptual framework for this study. Mistrust in professional health information is considered a potential perceived barrier to preventive engagement, while routine health behaviors, such as vaccination and regular check-ups, are indicative of health motivation and serve as cues to action. Applying the HBM provides a structured framework for understanding how cognitive appraisals and trust-related beliefs shape preventive behavior and influence sleep quality at the population level.

Therefore, this study aims to examine the associations between mistrust in professional health information, the use of preventive healthcare services, and sleep quality in a representative adult population.

Specifically, it seeks to address the following research questions:
Is higher mistrust toward professional health information associated with poorer sleep quality in adults?Does greater engagement in preventive health behavior correlate with better sleep quality?Do differences in preventive health behavior help explain the link between mistrust and sleep?


By addressing these questions, this study aims to advance the understanding of how cognitive and behavioral factors related to trust in health information influence sleep health at the population level.

## 2. Methods

### 2.1. Study Design, Setting, and Sample

This study is based on a cross-sectional, population-based survey conducted between 1 March and 30 May 2024 in the Autonomous Province of Bolzano (South Tyrol), Italy. The survey was collaboratively developed and implemented by the Provincial Institute of Statistics (ASTAT) and the Institute of General Practice and Public Health.

South Tyrol is located in northern Italy, bordering Austria, and is characterized by a multilingual population, with approximately 70% German speakers, 25% Italian speakers, and a minority speaking Ladin or other languages. This study targeted the resident adult population aged 18 years and older, estimated at around 400,000 individuals.

A stratified random sampling approach was employed to ensure representativeness across key demographic subgroups. Sampling strata were defined by age (18–34, 35–54, and ≥55 years), sex (male, female), citizenship (Italian or other), and municipality of residence. Using the provincial registry of residents, ASTAT randomly selected 4000 individuals to achieve adequate statistical power and precision, accounting for expected variation across strata. The aim of the sampling strategy was to yield a representative snapshot of the adult population of South Tyrol and to enable population-weighted analyses. Participation was voluntary, and data collection was carried out via a standardized bilingual questionnaire available in German and Italian.

### 2.2. Measures

#### 2.2.1. Sleep Quality

Sleep quality was assessed using the Brief Pittsburgh Sleep Quality Index (B-PSQI), a validated abbreviated form of the original Pittsburgh Sleep Quality Index (PSQI) [17]. The B-PSQI includes six items derived from the original 19-item scale, covering perceived sleep quality, sleep duration, sleep efficiency, sleep latency, and sleep disturbances. Each component score ranged from 0 to 3, with higher scores indicating poorer sleep. Component scores were summed to create a global B-PSQI score ranging from 0 to 15, with scores > 5 indicative of poor sleep quality [17,18]. Italian and German validated versions were used, showing good internal consistency (Cronbach’s α = 0.77 for Italian, α = 0.74 for German) [19,20].

#### 2.2.2. Mistrust Toward Professional Health Information

Mistrust in professional health information was operationalized using selected items from a broader questionnaire module that evaluated perceived trust in various health information sources [21]. Respondents were asked to indicate how much they trusted each source for health-related information, including outpatient specialists, family doctors, pharmacists, nurses, the Internet, books, personal feelings, and advice from friends or relatives. Responses were collected using a 4-point Likert scale ranging from 1 (“very”) to 4 (“not at all”). Higher numeric values corresponded to lower trust.

These items were adapted from previously used survey instruments [22] but have not been formally validated in their current configuration. A composite Mistrust Index was calculated by summing the responses to four professional sources: (1) family doctors, (2) outpatient specialists, (3) pharmacists, and (4) nurses. These sources were selected to represent trust in institutional, medically trained information providers. The resulting index ranged from 4 (very high trust) to 16 (very high mistrust). Due to the categorical scaling and absence of external validation, this index should be interpreted cautiously, and its use is limited to relative comparisons within the present study population.

#### 2.2.3. Preventive Health Behavior

Health-related behavior was measured using items adapted from the Health Behavior Checklist (HBC) [23], integrated as part of the lifestyle section of the questionnaire. Sixteen items captured routine health-related actions, such as maintaining physical activity, dietary practices, preventive medical and dental visits, hygiene, health information seeking, substance avoidance, and vaccination behavior [24]. Although all 16 items were included in the questionnaire, only a subset was selected for analysis in this study. These five items were chosen for their conceptual relevance to trust-sensitive health behaviors involving interaction with the healthcare system or professional information sources.

Respondents indicated how frequently they engaged in each behavior using a 4-point Likert scale: 1 = “never”, 2 = “rarely”, 3 = “often”, and 4 = “always.” This simplified scale was implemented for consistency within the overall questionnaire and deviates from the original 5-point scaling used in most HBC applications [23,24]. Therefore, official HBC scoring procedures and external comparison norms are not applicable.

The following five items were analyzed in the context of mistrust and sleep quality:“I let myself be vaccinated”.“I go regularly to the doctor for a check-up”.“I go regularly to the dentist for preventive check-ups”.“I discuss health issues with friends, neighbors, or relatives”.“I collect information about things that concern my health”.

Lifestyle-related items such as physical activity, diet, and substance use were not analyzed in this context, as their association with mistrust is likely to be indirect. The vaccination item was retained despite potential post-COVID sensitivity, due to its central role as a trust-driven behavior. Items were analyzed as ordinal variables. Depending on the analysis, responses were treated as continuous ordinal predictors in regression models (1–4 scale) or dichotomized (e.g., “often/always” vs. “rarely/never”) for categorical comparisons or mediation models.

No composite HBC score was generated. Analyses were performed on item-level associations with sleep quality and interactions with mistrust in health information, adjusted for sociodemographic variables and chronic disease status where applicable.

#### 2.2.4. Sociodemographic and Health Variables

Collected variables included age, gender, native language (German, Italian, and Ladin/Other), citizenship (Italy/Other), educational attainment (primary, vocational, high school, and university), urban versus rural residence, living situation (alone, with partner/family), self-rated health status (continuous, scaled 0–100), and self-reported presence of chronic disease(s) (binary: yes/no). Chronic disease burden was assessed using a checklist of predefined conditions (e.g., cardiovascular, pulmonary, metabolic, hepatic, oncological, nephrological, immunological, or mental illness). Respondents could indicate the presence of each condition, and the total number of reported conditions was summed to yield a count variable reflecting overall chronic disease burden.

### 2.3. Statistical Analysis

All analyses were conducted in Python version 3.10.12 using the cloud-based Google Colab (Google LLC, Mountain View, CA, USA) environment. Python scripts were developed with the support of OpenAI’s ChatGPT (GPT-4 model, OpenAI, San Francisco, CA, USA) and executed interactively in Jupyter-style notebooks. The analysis workflow combined several open-source packages, including pandas v2.2.2 (pandas development team) for data manipulation [25], numpy v1.26.4 (NumFOCUS, Austin, TX, USA) for numerical operations [26], scipy v1.13.0 (NumFOCUS) for statistical tests [27], and statsmodels v0.14.0 for regression modeling and diagnostics [28] and scikit-learn v1.4.2 (INRIA, Paris, France) for resampling procedures [29]. Data preparation, including the construction of the Mistrust Index and chronic illness indicators, was implemented in Python (Python Software Foundation, Wilmington, DE, USA). All scripts and the annotated dataset are available via the accompanying GitHub (GitHub, Inc., San Francisco, CA, USA) repository to ensure reproducibility.

Descriptive statistics were used to summarize sociodemographic, behavioral, and health-related variables, with means, standard deviations (SD), medians, interquartile ranges (IQRs), and proportions reported where appropriate. All analyses were weighted to reflect the demographic structure of the South Tyrolean population. Group differences across levels of mistrust and preventive behavior were tested using weighted chi-square tests for categorical variables. Relationships between continuous or ordinal variables, such as mistrust scores, preventive behavior frequencies, and sleep quality (B-PSQI), were analyzed using Spearman’s rank correlation.

To investigate associations between mistrust and sleep quality, as well as between preventive behavior and sleep quality, multivariable linear regression models were estimated. All models were adjusted for age (treated as a continuous variable), gender, education level, and the presence of chronic illness. To explore whether chronic disease modified these relationships, analyses were repeatedly stratified by chronic illness status. The linear regression assumptions of linearity, homoscedasticity, and the absence of multicollinearity were assessed using residual plots and variance inflation factors (a VIF < 2 was considered acceptable). To account for the non-normal distribution of the B-PSQI outcome, robust linear regression models using Huber’s M-estimator were computed as a sensitivity analysis.

Where relevant, analyses were conducted with and without the application of post-stratification weights to evaluate the effect of weighting on statistical inference. Associations are reported as standardized regression coefficients (β), Spearman’s ρ, and coefficients of determination (R^2^) to reflect the strength and explanatory power of the models. All reported models were adjusted for age, gender, education, and chronic illness.

## 3. Results

### 3.1. Sample Characteristics

Table 1 presents the key characteristics of the study population before and after the application of population weights. Overall, the weighting procedure led to minor adjustments in sociodemographic composition, notably correcting a slight overrepresentation of older individuals and women in the unweighted sample. The distributions of educational attainment, language groups, and municipality type also shifted modestly after weighting, aligning more closely with the official population structure. Preventive health behaviors, mistrust in professional health information, sleep quality, and the burden of chronic disease were broadly comparable between weighted and unweighted samples. While most differences were small, several reached statistical significance due to the large sample size.

### 3.2. Sociodemographic Variation in Mistrust Toward Professional Health Information and Preventive Health Behavior

In the weighted analysis, mistrust toward professional health information did not vary significantly across sociodemographic subgroups (Table 2). Age, gender, education, language group, municipality, and chronic disease showed comparable distributions across low, medium, and high mistrust levels, with no statistically significant differences observed.

Preventive health behaviors differed significantly across mistrust levels in the population (Table 3). For all five behaviors—vaccination, regular medical check-ups, dental visits, health information seeking, and health discussion—the distribution of response frequencies varied by mistrust group, with all *p*-values reaching statistical significance.

Across behaviors, individuals with high mistrust were consistently more likely to report lower engagement (e.g., “never” or “sometimes”), while those with low mistrust more often reported frequent or regular engagement (e.g., “often” or “always”). The association was most pronounced for vaccination and regular check-ups, where high mistrust was linked to markedly lower levels of reported preventive behavior.

These findings suggest that mistrust in professional health information is meaningfully associated with the likelihood of engaging in a range of preventive health behaviors in the general population.

### 3.3. Association Between Mistrust in Professional Health Information and Sleep Quality

Sleep quality tended to worsen with increasing levels of mistrust in professional health information. Although this pattern was observable across mistrust tertiles, the overall group difference did not reach statistical significance in nonparametric testing. Correlation analysis supported a weak positive association between mistrust and poorer sleep (Table 4).

To adjust for potential confounding, multivariable linear regression was conducted, confirming a statistically significant independent relationship between higher mistrust and poorer sleep quality after accounting for age, gender, education, and chronic disease status. The model explained a modest proportion of the variance in sleep scores. A robust regression analysis using Huber’s M-estimator yielded results that were nearly identical to those from the standard regression model, reinforcing the stability of the observed association despite potential violations of normality assumptions (Table 4). Other covariates included in the regression model (age, gender, education level, and presence of chronic illness) were not significantly associated with sleep quality and are therefore not shown in the table.

### 3.4. Association Between Preventive Health Behavior and Sleep Quality

Selected preventive health behaviors, including vaccination, routine check-ups, dental visits, health information seeking, and health discussions, were not meaningfully correlated with sleep quality in this sample (Table 5). Although weak bivariate associations were observed for some items, none reached statistical significance in adjusted linear regression models.

To further test the robustness of these findings, regression models were repeated using Huber’s M-estimator. This analysis confirmed the absence of significant associations across all behavioral indicators.

Although none of the selected preventive health behaviors showed significant associations with sleep quality in adjusted regression models, these null findings are reported in detail to demonstrate that preventive engagement does not explain the observed relationship between mistrust and sleep. These null findings support the interpretation that preventive behavior does not explain the association between mistrust and sleep quality.

## 4. Discussion

This study examined associations between mistrust in professional health information, preventive health behavior, and sleep quality in a representative adult population. Higher levels of mistrust were found to be associated with poorer sleep quality, as well as lower engagement in key preventive behaviors such as vaccination, medical check-ups, dental care, and health-related information seeking. In addition, when weighted analyses were applied, no significant differences in mistrust levels were observed across sociodemographic groups, including age, gender, education, language, or urban–rural residence. Although preventive health behaviors were less common among individuals with high mistrust, these behaviors did not appear to explain differences in sleep quality. This supports the interpretation that mistrust itself may act as a direct psychosocial factor.

The observed association between higher mistrust in professional health information and poorer sleep quality aligns with existing literature suggesting that cognitive and emotional stressors, such as skepticism toward health authorities or perceived informational unreliability, may contribute to sleep disturbances [31]. While mistrust has previously been linked to the lower utilization of preventive services and poorer health outcomes [32], the present findings suggest that this pathway may not operate indirectly through behavior changes in relation to sleep. The absence of significant mediation effects, despite strong associations between mistrust and lower preventive behavioral engagement, may indicate that mistrust exerts a more immediate psychosocial influence on sleep, potentially through mechanisms such as rumination, heightened threat perception, or reduced psychological safety [33]. These processes are known to heighten physiological arousal and cognitive hyperactivity [34], thereby contributing to difficulties falling or staying asleep. These results contrast with prior assumptions that behavioral disengagement explains health disparities associated with mistrust and instead underscore the need to consider mistrust itself a direct determinant of subjective well-being and health-related functioning [5].

While preventive behavior did not mediate the observed relationship between mistrust and sleep quality, other psychosocial mechanisms may plausibly explain this association. Mistrust may serve as a chronic cognitive stressor, heightening vigilance, increasing emotional reactivity, and reducing an individual’s sense of control or safety. These processes are known to disrupt sleep by promoting rumination, hyperarousal, and emotional dysregulation, even in the absence of behavioral disengagement. Recent research has also demonstrated that lower institutional trust is correlated with increased sleep fragmentation and heightened inflammation, thereby supporting a physiological connection between psychosocial mistrust and sleep disruption [35].

The findings highlight mistrust in professional health information as a direct and potentially modifiable risk factor for sleep disturbances, independent of preventive health behavior. The magnitude of the association between mistrust and sleep quality, while statistically significant, was small (β = 0.09), suggesting that additional factors likely contribute to variations in sleep outcomes. Public health efforts aimed at improving sleep and population well-being may benefit from targeting the informational environment and its trustworthiness, rather than focusing solely on individual behavior change [36]. Restoring trust requires more than corrective communication: it involves consistent transparency, the acknowledgment of past failures or uncertainty, and sustained engagement with communities [37]. Strategies such as culturally and linguistically adapted messaging, respectful dialog with marginalized groups, and the active inclusion of trusted intermediaries, such as general practitioners, pharmacists, or community leaders, may help re-establish credibility and reduce the psychosocial stress linked to mistrust [38]. In addition, sleep health promotion campaigns should consider addressing institutional and interpersonal trust as part of broader health literacy and preventive care strategies. These approaches may be particularly relevant in settings where behavioral disengagement stems not from access barriers alone, but from deeper credibility concerns.

This study has several methodological strengths, including its use of a population-based, post-stratified sample with linguistic and regional diversity, and the application of weighted analyses to ensure representativeness. Validated instruments were used to assess sleep quality and health behavior, and analytic models incorporated robust regression techniques to account for non-normal outcome distributions. The inclusion of covariate adjustment and sensitivity testing further supports the internal validity of the findings.

However, several limitations must be acknowledged. The cross-sectional design precludes causal inference and limits the ability to determine the temporal sequence between mistrust, behavior, and sleep quality. All data were self-reported, introducing the potential for recall or social desirability bias. Moreover, the preventive behavior items were adapted from the HBC using a 4-point scale, which limits comparability with other studies and may have constrained variance in mediation pathways. Moreover, the composite Mistrust Index employed in this study was newly developed from four items that assess trust in institutional health professionals. While the index demonstrated acceptable internal consistency, it has not undergone external validation and should therefore be interpreted with caution. Additionally, the use of tertile groupings, although beneficial for descriptive comparisons, may restrict comparability with other studies and obscure more nuanced variations in trust. Finally, data collection was conducted from March to May 2024, coinciding with the spring season in the Northern Hemisphere. Seasonal variations, such as increased daylight hours and milder temperatures, may affect both the duration and quality of sleep [39]. Consequently, the findings may not be entirely generalizable to periods characterized by more extreme environmental or behavioral conditions, such as winter or summer. Future research should take these seasonal effects into account.

## 5. Conclusions

This study demonstrates that mistrust in professional health information is independently associated with poorer sleep quality in a general adult population. While mistrust also correlates with reduced engagement in preventive health behaviors, these behaviors do not mediate the relationship with sleep. The results underscore the relevance of mistrust as a distinct psychosocial factor affecting health-related well-being and highlight the need for targeted strategies to build and sustain public trust in health communication and care systems.

## Figures and Tables

**Figure 1 healthcare-13-01385-f001:**
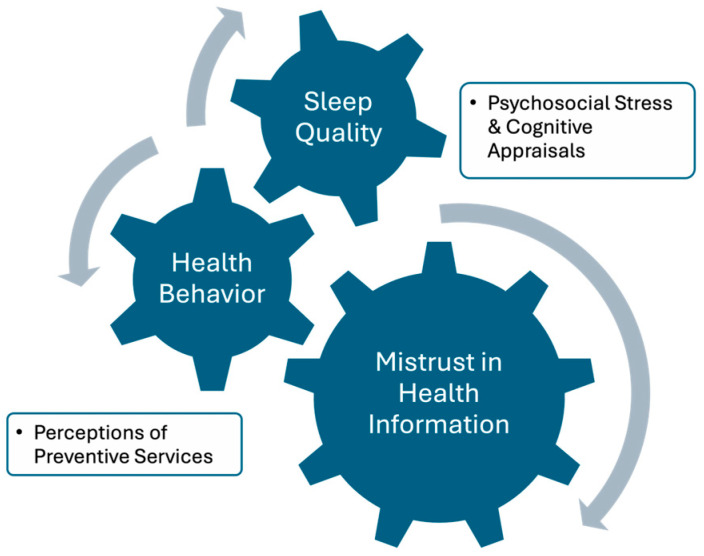
Conceptual framework of Health Belief Model pathways to sleep quality.

**Table 1 healthcare-13-01385-t001:** Weighted and unweighted descriptive characteristics of the study population, including sociodemographics, health behavior, mistrust, and sleep quality.

Variable	Unweighted Value ^1^	Weighted Value ^1^	*p*-Value ^7^
Sample size	2090	2090	—
Age (mean ± SD)	53.6 ± 17.8	50.9 ± 17.9	<0.001
Female (%)	55.2%	50.9%	0.006
Education			0.133
Middle or vocational school	55.8%	53.0%	—
High school	25.4%	26.1%	—
University	18.9%	20.9%	—
Language			<0.001
German	66.9%	63.7%	—
Italian	23.9%	23.0%	—
Other	9.2%	13.4%	—
Municipality ^2^			0.094
Urban	18.2%	20.3%	—
Rural	81.8%	79.7%	—
Mistrust Index (mean ± SD) ^3^	7.29 ± 2.09	7.32 ± 2.13	0.573
B-PSQI score (mean ± SD)	4.10 ± 2.94	3.99 ± 2.90	0.084
Poor sleep (PSQI > 5) (%)^4^	27.4%	25.7%	0.228
Chronic disease (%) ^5^	38.0%	35.1%	0.051
HBC Items ^6^			
See dentist, 4 (mean ± SD)	2.97 ± 0.98	2.92 ± 0.98	0.016
Gather information, 7 (mean ± SD)	2.59 ± 0.85	2.59 ± 0.85	0.701
Regular check-ups, 10 (mean ± SD)	2.56 ± 0.88	2.51 ± 0.88	0.016
Discuss health, 12 (mean ± SD)	2.54 ± 0.77	2.53 ± 0.78	0.596
Get vaccinations, 15 (mean ± SD)	2.88 ± 0.97	2.85 ± 0.97	0.149

^1^ Unweighted values reflect raw survey data; weighted values reflect post-stratification based on age, gender, and citizenship using official population data from ASTAT [30]. ^2^ Municipality distinguishes residents of the urban municipality of Bolzano from those in all other areas of the province. ^3^ Mistrust Index: higher scores indicate greater mistrust toward professional health information. ^4^ “Poor sleep” is defined as a B-PSQI score > 5. ^5^ Chronic disease is coded positive if any condition from a predefined checklist was self-reported. ^6^ HBC items refer to selected preventive behaviors, assessed on a 4-point Likert scale (1 = never, 4 = always). ^7^ *p*-values reflect weighted group comparisons using *t*-tests (continuous variables) or chi-square tests (categorical variables) based on the overall distribution. Abbreviations: B-PSQI, Brief Pittsburgh Sleep Quality Index; HBC, Health Behavior Checklist; and SD, standard deviation.

**Table 2 healthcare-13-01385-t002:** Weighted distribution of sociodemographic characteristics across mistrust levels in a representative adult population.

Variable	Total, *n*	Mistrust Toward Professional Health Information ^1^			*p*-Value ^3^
		Low	Medium	High	
Age Group					0.073
18–34	495	168 (34.0%) ^2^	217 (43.8%)	109 (22.1%)	
35–54	683	217 (31.9%)	305 (44.8%)	159 (23.3%)	
55–74	684	237 (34.6%)	286 (41.9%)	160 (23.5%)	
75	227	94 (41.4%)	99 (43.8%)	33 (14.9%)	
Gender					0.226
Female	1064	381 (35.8%)	459 (43.2%)	222 (20.9%)	
Male	1026	336 (32.8%)	449 (43.8%)	240 (23.5%)	
Education					0.390
High school	545	199 (36.5%)	239 (43.9%)	106 (19.6%)	
Middle or vocational school	1107	364 (32.9%)	487 (44.0%)	255 (23.1%)	
University	438	153 (35.2%)	182 (41.8%)	100 (23.1%)	
Language					0.798
German	1330	459 (34.5%)	580 (43.6%)	290 (21.9%)	
Italian	480	165 (34.5%)	212 (44.2%)	102 (21.3%)	
Other	280	92 (33.2%)	116 (41.8%)	70 (25.1%)	
Municipality					0.778
Rural	1665	565 (34.0%)	730 (43.8%)	369 (22.2%)	
Urban	425	151 (35.7%)	179 (42.2%)	93 (22.1%)	
Chronic Illness					0.478
Yes	733	257	320	156	
No	1357	460	588	309	

^1^ Mistrust levels were based on tertiles of the Mistrust Index (range: 4–16), with higher values reflecting greater mistrust in professional health information. ^2^ Percentages are row-wise and indicate the distribution of each sociodemographic subgroup across the three levels of mistrust (low, medium, and high). ^3^ Chi-square tests were computed using a weighted approximation of expected frequencies and are shown for each variable group.

**Table 3 healthcare-13-01385-t003:** Weighted distribution of preventive health behaviors across mistrust levels in a representative adult population.

Preventive Health Behavior ^1^	Total, *n*	Mistrust Toward Professional Health Information ^2^			*p*-Value ^4^
		Low	Medium	High	
Vaccination					<0.001
Never	168	41 (24.5%) ^3^	63 (37.5%)	64 (38.0%)	
Sometimes	662	160 (24.2%)	316 (47.7%)	186 (28.1%)	
Often	570	210 (36.8%)	259 (45.5%)	101 (17.7%)	
Always	690	306 (44.4%)	271 (39.3%)	113 (16.3%)	
Check-Ups					<0.001
Never	206	61 (29.4%)	80 (38.6%)	66 (32.0%)	
Sometimes	944	286 (30.3%)	431 (45.7%)	226 (24.0%)	
Often	602	224 (37.2%)	273 (45.2%)	106 (17.6%)	
Always	338	146 (43.4%)	126 (37.3%)	65 (19.3%)	
Dentist					<0.001
Never	177	73 (41.2%)	57 (32.5%)	46 (26.3%)	
Sometimes	561	164 (29.3%)	251 (44.8%)	145 (25.9%)	
Often	603	183 (30.3%)	281 (46.7%)	139 (23.1%)	
Always	750	297 (39.7%)	320 (42.6%)	133 (17.7%)	
Information Seeking					0.004
Never	229	84 (36.5%)	82 (35.7%)	64 (27.8%)	
Sometimes	686	213 (31.0%)	317 (46.2%)	156 (22.8%)	
Often	896	308 (34.4%)	409 (45.6%)	179 (20.0%)	
Always	280	113 (40.4%)	102 (36.6%)	64 (23.0%)	
Health Discussion					<0.001
Never	192	57 (29.7%)	76 (39.4%)	59 (30.9%)	
Sometimes	768	234 (30.5%)	352 (45.9%)	181 (23.6%)	
Often	952	351 (36.9%)	421 (44.2%)	180 (18.9%)	
Always	178	75 (42.0%)	61 (34.1%)	43 (23.9%)	

^1^ Responses are based on selected items from the Health Behavior Checklist (HBC), each assessed on a 4-point Likert scale. ^2^ Mistrust levels are based on tertiles of the Mistrust Index (range: 4–16), with higher values indicating greater mistrust toward professional health information. ^3^ Percentages are row-wise and represent the distribution of each response category across mistrust tertiles. ^4^ Chi-square test.

**Table 4 healthcare-13-01385-t004:** Associations between mistrust in professional health information and sleep quality (B-PSQI).

Statistic/Model	*n*	Estimate	95% CI	*p*-Value	Model R^2^
Low mistrust ^1^ (B-PSQI, mean ± SD) ^2^	724	3.92 ± 2.9	—	—	—
Medium mistrust (B-PSQI mean ± SD)	911	4.14 ± 2.87	—	—	—
High mistrust (B-PSQI mean ± SD)	455	4.29 ± 3.12	—	—	—
Kruskal–Wallis test	—	H = 4.99	—	0.083	—
Spearman correlation (Mistrust ~ B-PSQI)	—	ρ = 0.05	—	0.017	—
Linear regression (adjusted) ^3^	2090	β = 0.09	[0.03, 0.15]	0.003	0.063
Robust regression (adjusted, Huber’s M-estimator) ^4^	2090	β = 0.09	[0.03, 0.14]	0.003	—

^1^ Mistrust levels were categorized into tertiles based on the Mistrust Index (range: 4–16). ^2^ Higher B-PSQI scores indicate poorer sleep quality. ^3^ Linear regression was adjusted for age, gender, education, and chronic illness. ^4^ Robust regression using Huber’s M-estimator was applied to address potential non-normality. Abbreviations: B-PSQI, Brief Pittsburgh Sleep Quality Index; CI, confidence interval; and SD, standard deviation.

**Table 5 healthcare-13-01385-t005:** Associations between selected preventive health behaviors and sleep quality (B-PSQI).

Behavior Variable ^1^	Correlation ^2^		Linear Regression (OLS) ^3^			Linear Regression (Robust) ^3^		
	ρ	*p*-Value	β	95% CI	*p*-Value	β	95% CI	*p*-Value
Vaccination	0.04	0.1	0.01	[−0.12, 0.14]	0.886	0.02	[−0.10, 0.15]	0.724
Doctor check-ups	0.09	0.0	0.05	[−0.10, 0.20]	0.503	0.02	[−0.12, 0.17]	0.739
Dental check-ups	−0.02	275	−0.07	[−0.19, 0.06]	0.319	−0.07	[−0.20, 0.05]	0.253
Information seeking	0.03	145	0.07	[−0.08, 0.22]	0.338	0.05	[−0.09, 0.20]	0.480
Discussing health	0.03	0.13	−0.01	[−0.18, 0.15]	0.864	0.04	[−0.12, 0.20]	0.589

^1^ All behavior variables were derived from selected items in the Health Behavior Checklist (HBC), each measured on a 4-point ordinal scale (1 = never to 4 = always). ^2^ Spearman’s correlations represent bivariate associations between each behavior and the B-PSQI score. ^3^ Linear regression models were adjusted for age, gender, education level, and chronic disease. Robust regression analyses using Huber’s M-estimator were conducted to confirm model stability under potential violations of normality assumptions. Abbreviations: B-PSQI, Brief Pittsburgh Sleep Quality Index; OLS, ordinary least squares; β, regression coefficient; CI, confidence interval; and ρ, Spearman’s correlation coefficient.

## Data Availability

The original data presented in the study are openly available in GitHub (GitHub, Inc., San Francisco, CA, USA) at https://github.com/wiedermc/health-mistrust-sleep-analysis.git.

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
