# Peer review of "Health Information Mistrust Is Directly Associated with Poor Sleep Quality: Evidence from a Population-Based Study"

_healthcare, 2025, doi:10.3390/healthcare13121385_

Round 1

Reviewer 1 Report

Comments and Suggestions for Authors

Dear Author, 

 Thank you for this well-designed study examining the association between mistrust in professional health information, preventative health behaviours, and sleep quality. Below is a concise summary of my evaluation and actionable recommendations.

 Recommendations for Revision:

Introduction: Define mistrust operationally early on (currently appears after HBM discussion).

Results:

Table 1: Include p-values for all weighted vs. unweighted comparisons (currently missing for some rows). 

Discussion:

 Expand on how mistrust might directly impair sleep (e.g., rumination, hypervigilance). 

Abbreviations:  

Define all abbreviations at first use (e.g., "HBM" appears before definition). 

Best regards,

Author Response

Comment 1: Define mistrust operationally early on (currently appears after HBM discussion).

We agree and have revised the Introduction to include an earlier operational definition of mistrust in professional health information. A sentence was added immediately after introducing the topic of psychosocial influences on sleep quality. The definition was drawn from our Methods section and describes mistrust as a generalized skepticism toward medical professionals and institutions and includes an additional reference.

Following text was updated: "Mistrust in professional health information refers to a general skepticism toward healthcare providers, institutions, and their recommendations, and has been linked to reduced engagement with preventive services and poorer health outcomes [5]."

Comment 2:  Table 1: Include p-values for all weighted vs. unweighted comparisons (currently missing for some rows).

We would like to clarify that p-values in Table 1 are presented at the variable level for categorical variables (e.g., education, language, municipality), based on overall weighted chi-square tests. Individual response categories (e.g., "high school", "university") are part of the same variable and therefore do not have separate p-values reported per row. This is consistent with standard practice for reporting categorical comparisons. To maintain clarity and avoid redundancy, we have not added additional row-level p-values. However, we revised the table note [7] to better explain that p-values refer to the overall distribution of each variable, not individual categories. We hope this addresses the concern.

Change in manuscript: "p-values reflect weighted group comparisons using t-tests (for continuous variables) or chi-square tests (for categorical variables, based on the overall distribution)."

Comment 3: Expand on how mistrust might directly impair sleep (e.g., rumination, hypervigilance).

We have expanded the Discussion to further elaborate on potential mechanisms linking mistrust to poor sleep quality. These include cognitive-emotional pathways such as rumination, perceived threat, and reduced psychological safety, which may lead to hypervigilance and difficulty initiating or maintaining sleep.

Change in manuscript: "Mistrust may act as a psychosocial stressor that directly impairs sleep by promoting rumination, increasing threat perception, and reducing feelings of safety and control. These processes are known to heighten physiological arousal and cognitive hyperactivity [31], thereby contributing to difficulties falling or staying asleep."

Comment: Define all abbreviations at first use (e.g., "HBM" appears before definition).

We have reviewed the manuscript and ensured that all abbreviations, including HBM (Health Belief Model), are defined at their first mention in the text.

Reviewer 2 Report

Comments and Suggestions for Authors

Dear Editor,

I would like to thank the authors of the manuscript ID healthcare-3663678 entitled "Health Information Mistrust Is Directly Associated with Poor Sleep Quality: Evidence from a Population-Based Study" for presenting the results of their study on the effects of the professional health information mistrust effects on subjective sleep quality at the population level, and preventive health behavior mediation effects.

The manuscript presents the results of a cross-sectional analysis of a representative adult sample (n ≈ 2,100) from South Tyrol, Italy. The authors based their study concept on the Health Belief Model (HBM) and used data collected from questionnaires on mistrust toward professional health information (creating a composite Mistrust Index), five preventive health behaviors (drawn from Health Behavior Checklist, HBC), and subjective sleep quality (Brief Pittsburgh Sleep Quality Index, B-PSQI).

The authors used basic statistical analyses, as well as multivariable linear regression, robust regression (Huber’s M-estimator), and nonparametric correlation in order to assess possible association between mistrust, behavior, and subjective sleep quality.

After reading this article in details, my main impression is that the article is very well written, in adherence to Journal's standards, and addresses a very important issue on the effects of mistrust in professional health information on presence and progression of sleep disorders, as well as the effect of preventive health behaviors on good sleep. Sleep disorders are very prevalent in the adult population and if left unrecognized or untreated can lead to serious consequences on patients' health and quality of sleep and life.

There are several issues for consideration:

  1. Title: Adequate.
  2. Abstract: Adequate
  3. Introduction: Adequate.
  4. Methods: Adequate. Remarks:
  • The authors did not offer an explanation why elements of HBC such as maintaining physical activity, dietary practices, substance avoidance etc. were not considered important preventive engagement relevant to sleep and trust-related pathways…while the vaccination question, in the post COVID era carries a lot of potential bias.
  • This simplified scale was implemented for consistency within the overall questionnaire and deviates from the original 5-point scaling used in most HBC applications (REF). Missing the reference
  • Items will be analyzed as ordinal variables. Depending on the analysis, responses may be treated as continuous ordinal predictors in regression models (1–4 scale), or dichotomized (e.g., "often/always" vs. "rarely/never") for categorical comparisons or mediation models. Items were analyzed, not will be. Responses were treated as….

  1. Results: Adequate. Remarks:
  • The effect of chronic illnesses was not presented, table 1 mentions the difference between unweighted and the weighted sample, with approximately one third of the sample reports chronic illnesses, but they are not addressed in table 2, where it could be expected that subjects in the higher need of health services have more defined opinion on professional health information quality. So this category of subjects should be presented in table 2, as well as the association with preventive health behavior

  1. Discussion: Adequate. Remarks:
  • Limitations should, as in Methods, mention the limited usability of a composite Mistrust index that was created specifically for this study, and presented in tertiles
  1. References: Adequate, relevant.
  2. Tables and figures: Adequate, relevant.

Reviewer 3 Report

Comments and Suggestions for Authors

This manuscript addresses an important and timely topic: the relationship between mistrust in professional health information and sleep quality. The authors present a well-structured, population-based study grounded in a conceptual framework (Health Belief Model), and the methodological approach is generally rigorous. The findings offer a novel contribution to public health literature, emphasizing mistrust as a direct psychosocial risk factor for poor sleep, independently of preventive behavior.

The manuscript is clearly written, the methodology is well-articulated, and the statistical analyses are robust, including sensitivity testing. The interpretation is cautious and nuanced, and the implications for public health practice are thoughtfully discussed. However, I suggest several revisions to strengthen further the clarity, robustness, and theoretical coherence of the manuscript.

-The manuscript concludes that preventive behaviors do not mediate the relationship between mistrust and sleep quality, but it does not sufficiently explore alternative pathways.

-I recommend expanding the discussion to speculate on potential mechanisms (e.g., stress, perceived uncertainty, rumination, emotional dysregulation) and how mistrust could lead to sleep disturbance outside of behavioral disengagement, i suggest mentioning https://pubmed.ncbi.nlm.nih.gov/40015959/.

-While the association between mistrust and sleep quality is statistically significant (β = 0.09), the effect size is small. Consider adding commentary on clinical relevance and whether the observed effect has practical implications for public health interventions.

-While the HBM is mentioned as a guiding framework, it is not operationalized in the analysis. I suggest briefly noting which components (e.g., perceived barriers, susceptibility) align with the mistrust and preventive behavior constructs, or clarifying whether the model is used only for conceptual framing.

-Data were collected between March and May 2024. Sleep patterns may be seasonally influenced. Please discuss whether seasonality could have impacted results and how this might limit generalizability.
